# Surfactant and Block Copolymer Nanostructures: From Design and Development to Nanomedicine Preclinical Studies

**DOI:** 10.3390/pharmaceutics15020501

**Published:** 2023-02-02

**Authors:** Orestis Kontogiannis, Dimitrios Selianitis, Nefeli Lagopati, Natassa Pippa, Stergios Pispas, Maria Gazouli

**Affiliations:** 1Laboratory of Biology, Department of Basic Medical Science, School of Medicine, National and Kapodistrian University of Athens, 11527 Athens, Greece; 2Theoretical and Physical Chemistry Institute, National Hellenic Research Foundation, 48 Vassileos Constantinou Avenue, 11635 Athens, Greece; 3Department of Pharmaceutical Technology, Faculty of Pharmacy, Panepistimioupolis Zographou, National and Kapodistrian University of Athens, 15771 Athens, Greece

**Keywords:** nanomedicine, block copolymer, surfactant, drug delivery, pharmacokinetics

## Abstract

The medical application of nanotechnology in the field of drug delivery has so far exhibited many efforts in treating simple to extremely complicated and life-threatening human conditions, with multiple products already existing in the market. A plethora of innovative drug delivery carriers, using polymers, surfactants and the combination of the above, have been developed and tested pre-clinically, offering great advantages in terms of targeted drug delivery, low toxicity and immune system activation, cellular biomimicry and enhanced pharmacokinetic properties. Furthermore, such artificial systems can be tailor-made with respect to each therapeutic protocol and disease type falling under the scope of personalized medicine. The simultaneous delivery of multiple therapeutic entities of different nature, such as genes and drugs, can be achieved, while novel technologies can offer systems with multiple modalities often combining therapy with diagnosis. In this review, we present prominent, innovative and state-of-the-art scientific efforts on the applications of surfactant-based, polymer-based, and mixed surfactant-polymer nanoparticle drug formulations intended for use in the medical field and in drug delivery. The materials used, formulation steps, nature, properties, physicochemical characteristics, characterization techniques and pharmacokinetic behavior of those systems, are presented extensively in the length of this work. The material presented is focused on research projects that are currently in the developmental, pre-clinical stage.

## 1. Introduction

Polymers hold a wide range of applications in medicine in terms of producing and developing novel pharmaceutical compounds against a wide range of diseases, especially as components of drug delivery systems, with their synthesis protocols being well established in the laboratory process [1,2]. The formulation of block copolymer nanosystems, belonging to the wider category of materials characterized as “soft materials”, is a more time-consuming process that results in a finished product with a higher cost, when compared to polymeric nanoparticles. Nevertheless, a great deal of research is being performed in the field with multiple products having already been approved, which exhibit superior ability to combat a wide range of disorders [1,3]. The use of copolymers in a mixture along with other amphiphilic species/compounds enables the formation of various nanostructures through a process called microphase-separation. Today, the use of environmentally safe block copolymers represents a novel family of multifunctional materials [4,5,6,7]. 

Molecules of amphiphilic nature usually contain two or more components, each with its own and possibly different affinity towards a particular solvent. The occurrence of highly organized systems via self-assembly is amongst the most common occurring phenomena in nature (e.g., the spontaneous self-assembly of amphiphilic biomolecules) and great efforts have been made in the duplication of these processes in artificial environments, through the exploitation of components that form different types of vesicles or other structures to minimize the free energy of the system. This ultimately takes place through the minimization of the surface area of the lipophilic component of the particle “available” to interact with water. Today, self-assembly is amongst the most prominent candidates for nanofabrication as it represents an easy and simple method to gain access to complex and diversified structures on the nanometer scale [5]. A wide range of vesicles originating from the mixture of low and high molecular weight amphiphiles, along with proper structural characteristics such as size, membrane width, and mechanical stability, can formulate nanoparticulate systems with great functionalization capabilities, which hold promise as drug delivery carriers for the encapsulation and release of multiple hydrophobic and hydrophilic pharmaceutical compounds, and even allow in certain cases the co-delivery of multiple drugs [8,9]. The unique characteristics of each amphiphile leads to the creation of highly complex, versatile and diversified mixed systems. The formulation of block copolymer nanoparticles, along with small molecular weight surfactants, offers a final formulation that has more adjustable parameters than most other binary organic systems (mostly due to greater structural variability between the individual components), while their highly versatile synthesis process offers the opportunity to easily create novel hybrid materials [8,9,10]. Many areas of interest in which such nanoparticulate systems may hold great promise include, but are not limited to, the pharmaceutical sector, environmental technologies development, industrial foaming, the cosmetics industry, drug solubilization, oil recovery, and as mediums for the formation of metal nanoparticle systems [4,11,12].

The aim of this review is to categorize, summarize, and present selected recent scientific results on the applications of surfactant-based, polymer-based and mixed nanoparticulate drug formulations intended for use in different ways in the medical field and in drug delivery. The nature, development process, overall properties, physicochemical characteristics, characterization techniques and pharmacokinetic behavior—both in vitro and in vivo—of those systems, are presented in detail in this work, as well as the promise that they hold in terms of future clinical translation. The approaches presented and discussed in this review are in the pre-clinical stage and are handpicked amongst the most reliable, state-of-the-art, and prominent works, published in esteemed scientific journals, and have all been through the peer review process. 

## 2. Methodology

Systematic search and review of papers regarding polymer-surfactants took place via MedLine, Scopus, and Web of Science platforms and abstract presentations of international conferences. The keywords that used were: block copolymers AND surfactants.

## 3. Surfactant Nanosystems

Human use of surfactants can be traced all the way back to products derived from plant oils and animal fat that have been used in hygiene as soap, detergent, foaming and cleaning agents [13]. Surfactants can come from both natural and synthetic sources and hold a wide range of promising applications in the pharmaceutical industry, being used for increasing the solubility of poorly dissolved agents and as drug delivery system components, towards the development of novel surfactant-based drug delivery platforms [14,15]. Other applications of such flexible chain compounds, mainly in the food and cosmetics industry—with emerging applications in a wide range of multiple diverse technologies—are part of the interest that surrounds these molecules, which enable the optimization of the physicochemical and stability characteristics of several products [16]. 

Surfactants are molecules that are amphiphilic in nature and consist of two distinct structural parts, a hydrophilic head group and a lipophilic tail (either single or double chain) and, depending on other characteristics such as the HLB ratio (hydrophilic to lipophilic balance) or their charge (ionic/non-ionic molecules) as well as their molecular weight, can be categorized into different groups [14,15,17]. The term surfactants refers to molecules that lower the interfacial tension between two or more molecular components of a material system and result in a formulation with improved compatibility, colloidal stability, and dispersion characteristics [13]. In the case of nanoparticulate systems, surfactants are widely studied and used, mostly as additives to previously examined particle systems that exhibit a level of phase incompatibility, for the creation of stable drug and gene delivery platforms with improved absorption and distribution profiles, as well as better delivery of poorly soluble (lipophilic) drug compounds, including multiple anticancer agents such as methotrexate and paclitaxel, by resulting in formulations that are better equipped to overcome a variety of different biological barriers [13,14,18]. 

The process of self-assembly of small molecular weight amphiphilic molecules (i.e., surfactants) has been thoroughly studied over the years, while a general classification has been made regarding the type of charge of the polar head group (Figure 1). In short, the three main categories are those of cationic (CTAB, CTAC), anionic (C14Na), and zwitterionic (dodecyldimethylamine oxide), and finally, a fourth category is that of non–ionic molecules (Tween 80, Span 80) [19].

Nanosystems containing surfactants have exhibited great promise in the areas of anticancer drug delivery, as well as in early detection and imaging, with novel applications being developed on a yearly basis, while their use also falls under the scope of personalized medicine, enabling the development of tailor-made, individualized therapies [14]. Entities with a higher hydrophilic to lipophilic balance (Table 1) give compounds that are more water soluble, and the binding of surfactants to nanoparticles gives the original particle certain hydrophobic or hydrophilic properties, enabling a better dispersion in the selected medium [13,14]. In terms of the binding process, grafting surfactants in different nanoparticles can occur in either covalent assembly or non-covalent absorption, with the binding resulting in changes in both entropy and enthalpy [13].

In cancer research, the addition of surfactants to nanoparticle delivery platforms has been shown to tackle, to a degree, several pre-existing problems relating to the use of more conventional drugs, including non-specific targeting, low therapeutic efficiency, and drug resistance, as well as several potential side-effects, relating mostly to the distribution and cellular absorption (in this case endocytosis) of large amounts of toxic anticancer compounds from healthy cells [14].

Lipophilic non-ionic surfactants can better solubilize poorly soluble compounds, with the additional advantage of being less toxic than their ionic counterparts, and are thus more preferable in today’s pharmaceutical industry as drug delivery platform components. Among the most notable non-ionic surfactants are Span 80, Tween 80, Tween 20, and Brij 97, and their addition to nanoparticles often results in changes in particle size, shape, and stability [17]. In some cases, the use of co-surfactants was observed to help with the stabilization of structurally weaker areas of the nanoparticle membrane/surface [17].

Another notable example of the use of surfactant-based drug delivery vesicles systems in cancer treatment is the preparation of non-ionic amphiphilic vesicles containing different mixtures of Span 20 and Tween 80 along with cholesterol for the delivery of doxorubicin against metastatic and non-metastatic breast cancer. Researchers concluded that almost all formulations, i.e., mixtures of the different surfactants, were able to incorporate significant amounts of doxorubicin, exhibiting satisfactory stability, with or without the API, and were able to achieve a steady release profile for 72 h after administration. In vitro experiments using the MCF-7 and MDA MB 468 cell lines reached the conclusion that, while formulations containing Span 20 were internalized to a higher degree and thus exhibited a more prominent anticancer effect, they were also the most cytotoxic when compared with a multicomponent formulation containing Tween 80 [20]. 

In terms of already existing products on the market, a great example is that of Taxotere©, which is considered amongst the most prominent chemotherapeutic drugs available today. It is lipophilic in nature and its active pharmaceutical ingredient is Docetaxel along with two excipients, ethanol and Tween 80 [21]. In another study, nanoparticles containing Docetaxel formulated with Poloxamer F127 (Table 2) copolymer exhibited a stronger hydrophobic interaction with the API in comparison to nanoparticles formulated with the surfactant of the Pluronic family F68 [21].

Niosomes are self-assembled, synthetic vesicles of nanoparticle size that are formulated through the hydration of non-ionic surfactants often with the combined use of cholesterol or other molecules of amphiphilic nature [18]. Exhibiting many similarities to liposomes regarding their potential use as drug delivery platforms, niosomes offer many stability and manufacturing advantages, including a lower formulation cost and easier upscaling of the production [15,26,27,28]. Surfactants that exhibit HLB ratio values in the range of 4 to 8 have been found to be more suitable for the formation of niosomes, while others such as sorbitan monooleate (Span 80, HLB of 4.3) are unable to formulate niosomes on their own due to their improper molecular geometry [26]. 

In previous studies, the use of alkyl glycerol ethers in niosome formulation has successfully altered the pharmacokinetic profiles of methotrexate and doxorubicin [26], while, more recently, the potential uses of other types of niosomal formulations such as discomes (large niosomes of 11–60 mm in size) and polyhedral niosomes (i.e., niosomes obtained from non-uniform surfactants) are being studied [26]. Another category exhibiting great promise is that of pronisomes which are vesicles of greater sizes, easy to produce, and with greater physical stability [29]. Last but not least, niosomes have been somewhat studied and have shown promise in certain pre-clinical applications regarding the delivery of active pharmaceutical compounds (APIs) through alternative means of administration (other than intravenous or intramuscular), such as oral, transdermal, pulmonary and ocular administration, thus being able to overcome the acidic environment and enzymatic degradation of the GI track [17]. In many cases, cholesterol is utilized as a stabilizing agent, increasing the overall drug bioavailability and the absorbed percentage of the drug, through altering the niosomal membrane stability and rigidity by occupying the space that would otherwise be void between the molecules that form the bilayer. It is of critical importance that the addition of cholesterol should be below a certain concentration, above which drug stability is compromised due to the cholesterol molecules antagonizing with surfactant molecules for space [15]. Other forms of niosomes, such as elastic niosomes, have been investigated to a lesser extent (Tween 61 and Span 60 niosomes entrapping diclofenac diethyammonium and embodying 0–25% ethanol) [26].

Lastly, larger surfactant molecules have been studied exhibiting properties that bridge the gap between small weight, conventional surfactants, and block copolymers in terms of their self-assembly behavior, and they are classified as “giant surfactants”. The main advantage offered by the utilization of such systems is the integration of the advantages of both worlds, i.e., of smaller and larger biocompatible amphiphiles, to the final formulation. Thus, they provide an innovative tool for the formulation of versatile nanoengineered drug delivery platforms with larger hydrodynamic radii yet still exhibiting the same sub-10 nm organizational properties and structural features which those conventional small molecules offer. Such molecules exhibit the ability to create microsystems of great versatility, which are composed of nano building blocks of high organization and which result in systems created in multiple states (bulk, thin film, solution) that can possibly be integrated into many novel applications [30].

## 4. Block Copolymer Nanosystems

Polymeric nanosystems have been used for many decades as innovative drug carrier platforms. Polymers have the ability to self-assemble in aqueous media (Figure 2) achieving dimensions at the nanoscale which have the potential to encapsulate hydrophobic drugs or other bioactive compounds [1,31]. In recent years, polymer research has focused on materials with different macromolecular architectures (i.e., linear, stars or hyperbranched copolymers) that respond to external stimuli such as pH [4,5,32,33], temperature [6,8], and ionic strength [9,34]. Self-organized polymeric nanoparticles (NPs) present a wide variety of properties and could be regarded as potential drug vehicles through control of their size, structure and morphology, in terms of a wide variety of functions, in vitro and in vivo [35,36,37].

Yang and co-workers reported novel dual targeting diblock copolymers consisting of poly(ethylene glycol) (PEG) and poly(ε-caprolactone) (PCL) having a tumor targeting ligand, folic acid, which were self-organized into micelles. The acquired micelles were capable of co-entrapping superparamagnetic iron oxide (SPIO) NPs and doxorubicin (DOX) forming hybrid NPs with 50 nm in hydrodynamic radius (R_h_). At 25 °C the hybrid NPs were superparamagnetic and converted to ferrimagnetic at 10 K. Drug release studies demonstrated that the SPIO-DOX-loaded micelles were able to release up to 70% of the drug at acidic conditions, whereas at physiological conditions only 10% of the drug was released. Furthermore, the SPIO-loaded and folate-functionalized micelles had a significant response when an external magnetic field was applied, resulting in more efficient transport into the tumor cells [38]. 

Another example of the formulation of a hybrid, polymer-based, micellar nanosystem intended for drug delivery is that composed of the novel amphiphilic block copolymers P(MMA-co-HPMA)-b-POEGMA, which contain monomers of differing polarity and in a variety of weight ratios, utilized for the encapsulation and transportation of curcumin and indomethacin. The copolymers were produced through the use of the RAFT technique, while again, the different hydrophilicity/lipophilicity of the individual monomeric units enabled them to self-assemble once exposed to aqueous solutions, resulting in amphiphilic nanoparticles with hydrophobic cores. In both cases of encapsulated APIs, the micellar aggregates retained their original characteristics and colloidal stability for at least ten days [39].

Zheng et al. developed a novel multifunctional nanosystem, consisting of poly (lactic-co-glycolic acid) (PLGA)-lecithin polyethylene glycol (PEG) core-shell NPs, that incorporated the remarkable characteristics of liposome and polymeric NPs for chemotherapeutics delivery. The physicochemical characteristics were easily adjusted by varying formulation parameters such as lipid/polymer ratio and modifying terminal groups of 1,2-distearoyl-sn-glycero-3-phosphoethanolamine (DSPE)-PEG. They utilized a hydrophilic drug cis-platin (DDP) or hydrophobic DDP prodrug in the core-shell NPS, and found large entrapment efficacy, significant stability and special FA targeting recognition for MCF-7 cells, with FA receptors overexpression and excellent cytotoxicity. The remarkable results obtained for these multifunctional NPs evidenced that these NPs could be used as drug delivery platforms for chemotherapy [40].

Another interesting approach to targeted and efficient delivery of therapeutic agents to cancer cells was reported by Sahoo et al [41]. In particular, they synthesized a multi-responsive poly(*N*-isopropylacrylamide)-block-poly-(acrylic acid) diblock copolymer by reversible addition-fragmentation chain transfer (RAFT) polymerization. The formed NPs consisting of the smart shell were coated with magnetic nanoparticles (MNPs), along with DOX as an anticancer drug agent and cancer cell-specific targeting agents. The modification of surface on MNPs introduced amine groups utilizing 3-aminopropyltriethoxysilane, resulting in their adhesion to the copolymer via EDC/NHS method. Moreover, to achieve cancer-specific targeting properties, folic acid was attached to the surface of the NPs. Afterwards, a fluorescent agent, namely rhodamine B isothiocyanate, was used as a fluorescent probe to the MNPs for cellular imaging applications. The NPs were investigated by a plethora of physicochemical characterization techniques. Furthermore, the authors utilized an anticancer drug DOX to evaluate the drug release profile under different conditions. DOX-loaded MNPs revealed that, at acidic conditions (pH 5.0) and at 37 °C, 75% of the loaded DOX was released, whereas 42–43% of the drug was released at pH 5.0 and 25 °C. Also, 23% of the drug was released at pH 7.4 at 37 °C. Furthermore, the biological features of the NPs were explored by MTT assay, fluorescence microscopy and apoptosis tests. In vitro apoptosis activity showed that the drug-loaded NPs caused noticeable death levels to the HeLa cells. These NPs could potentially be used as drug-loaded carriers for in vivo studies [41].

Chen et al. constructed a smart dually responsive polymeric system with a controllable drug release for the treatment of cancer. They synthesized an ultrasound and pH-responsive poly(ethylene oxide)-b-poly-(2-(diethylamino) ethyl methacrylate-stat-poly-(2-tetrahydrofuranyloxy)ethyl methacrylate [PEO-b-P(DEA-stat-TMA)] via atomic transfer radical polymerization (ATRP). This responsive copolymer self-assembled into vesicles in tetrahydrofuran (THF)/neutral water (pH 7.4) by dialysis method for removal of THF. The results from these smart polymer vesicles revealed no cytotoxicity below 250 μg/mL and successful encapsulation of the hydrophobic anticancer drug DOX. The release profile of loaded vesicles showed a sustained release of DOX when treated with ultrasound radiation or by changing the pH at 37 °C [42].

Du and co-workers synthesized pH-responsive degradable chimaeric polymersomes for active loading and controlled release of doxorubicin hydrochloride (DOX). Poly (ethylene glycol)-b-poly(2,4,6-trimethoxybenzylidene-1,1,1-tris(hydroxymethyl)ethane methacrylate)-b-poly(-acrylic acid) (PEG-b-PTTMA-b-PAA) triblock copolymers were synthesized via RAFT polymerization. PEG-b-PTTMA-b-PAA copolymers formed mono-dispersed polymersomes in nanoscale dimensions of around 63.9–112.1 nm. By using transmission electron microscopy (TEM) and confocal laser scanning microscopy (CLSM) the polymersome structure was determined. These polymersomes were able to successfully encapsulate a large amount of the drug. In vitro drug release profiles showed that drug-loaded chimeric polymersomes released DOX in a controlled and pH-dependent manner. CLSM studies indicated that these nanoparticles could successfully deliver and release DOX into the nuclei of HeLa cells. Furthermore, MTT assays in HeLa cells indicated that drug-loaded polymersomes showed significant anti-tumor action close to that of the free drug. The acquired data determined that this pH-responsive degradable polymersome system could be a useful drug vehicle for tumor therapy [43]. 

Hami and co-workers synthesized a new micellar drug delivery system which was pH-responsive. They prepared a poly(lactic acid)-b-poly (ethyleneglycol) (PLA-b-PEG) block copolymer via ring opening polymerization (ROP). The preparation of folate-conjugated block copolymer by utilizing an acid labile hydrazone linkage followed. Docetaxel (DTX) was used as a hydrophobic chemotherapeutic agent for encapsulation into the nontargeting PLA-b-PEG and targeting PLA-b-PEG-FOL block copolymers. The drug release profiles revealed that, under acidic conditions the maximum drug release was 80% whereas, under physiological conditions only 30% of the drug was released. Therefore, these results may be considered useful for further explorations of these pH-responsive nanosystems in biomedical applications [44].

Su et al. developed a novel thermo-responsive poly(N-isopropyl acrylamide)-poly(L-lactide)-poly(N-isopropyl acrylamide) (PNIPAAm-PLLA-PNIPAAm) triblock copolymer via atom transfer radical polymerization (ATRP). The self-assembly in aqueous media was studied by dynamic light scattering (DLS) and transmission electron microscopy (TEM). The triblock copolymers formed micelles with sizes between 20 and 40 nm. In vitro cytotoxicity assay was performed to evaluate the cytotoxicity of copolymers as well as the hemocompatibility, which was evaluated from hemolysis. The obtained results reveal that triblock copolymers possess significant biocompatibility which is ideal for biomedical applications [45].

Li et al. reported an interesting approach to synthesizing a pH-sensitive polypeptide-based nanogel system for drug delivery. The formed nanogels were constructed by using a hydrophilic methoxy poly(ethylene glycol)-b-poly[N-[N-(2-aminoethyl)-2-aminoethyl]-L-glutamate] (MPEG-b-PNLG) and hydrophobic terephthalaldehyde (TPA) as a crosslinker. Utilizing DLS it was possible to determine the sizes of nanogels prepared. The mean nanoparticle size was 56 nm at physiological conditions. DOX was used as a hydrophobic drug for its encapsulation in the hydrophobic core of the nanogels. Afterwards, the authors evaluated the capability of these nanogels to release the drug under different conditions. They found that, under acidic conditions, a faster release takes place than in a physiological environment. Moreover, this system exhibited an enhanced anticancer behavior against MDA-MB-231 cells compared to free DOX. The data on these pH-responsive nanogels could be useful for enhanced tumor therapy [46].

Wang’s group constructed a novel drug delivery system via radical polymerization based on chitosan (CS) and N-isopropylacrylamide (NIPAAm) with acrylamide (AAm), CS-poly(NIPAAm-co-Aam) resulting in nanogels. They utilized a model drug, namely paclitaxel (PTX), to encapsulate into formed nanogels. They studied the thermally responsive behavior from PTX-loaded nanogels and the biological properties of these. Drug release profiles revealed that the PTX-loaded nanogels were affected by temperature. It is worth mentioning that the drug release level significantly increased from about 15% at 25 °C and 32 °C to more than 30% at 38 °C and 39 °C. Moreover, they investigated the thermally responsive cellular uptake of these nanogels, by using a hydrophobic fluorescence probe called coumarin-6. They found that the intracellular fluorescence was enhanced with time and temperature. In vivo experiments revealed a higher antitumor efficacy of PTX-loaded nanogels compared to PTX solutions. This thermally responsive nanogel system could be a potential candidate for the combination of thermal therapy and chemotherapy [47].

An interesting work for cancer-targeted drug delivery and magnetic resonance imaging (MRI) was reported by Li et al [48]. They developed a novel amphiphilic multiarm star block copolymer system, H40-PCL-b-P(OEGMA-Gd-FA), which formed polymeric unimolecular micelles in aqueous media capable of encapsulating the hydrophobic drug paclitaxel (PTX). H40 was the fourth generation hyperbranched polyester core, PCL was the hydrophobic inner layer and poly(oligo(-ethylene glycol) monomethyl ether methacrylate) (POEGMA) was the hydrophilic outer corona which was covalently labeled with DOTA-Gd (Gd) and folic acid (FA) for synergistic targeted drug delivery and MR imaging. PTX-loaded unimolecular micelles revealed a sustained release profile of up to 80% loaded PTX after 5 days. In vitro cytotoxicity studies exhibit significantly higher cytotoxicity in contrast to bare unimolecular micelles. Furthermore, in vivo MR imaging studies in rats indicated an improved positive contrast and extended blood circulation time for these unimolecular micelles [48].

## 5. Surfactants-Block Copolymer Mixed Nanosystems

As an extension of the nanosystems discussed so far, surfactant-block copolymer mixed nanosystems have been used in the preparation of various mixed nanoparticles for drug delivery, being able to solubilize compounds of high hydrophobicity that, at the same time, exhibit low solubility and permeability toward cells, such as a variety of antineoplasmatic APIs like methotrexate and paclitaxel [18,49,50,51]. 

The formulation of a successful drug delivery system depends strongly on its ability to achieve temporal and distribution control, which indicates the extent to which the timing and duration of the drug release after administration, as well as the compartment (either a specific group of cells or a type of tissue or organ) at which the release takes place can be controlled. In various examples, the incorporation of low molecular weight surfactants into micellar block copolymer systems achieved a sustained release of the encapsulated API, indicating encouraging potential for solid tumor therapy. In such mixed nanosystems, binding occurs between the hydrophobic core of the block copolymer micelles and the surfactant hydrophobic tails, with surfactant heads being able to interact with the hydrophilic corona segments. The strength of the bond is dependent on the micelle’s hydrophobicity, the length of the hydrocarbon chain of the surfactant and the intrinsic characteristics of the head group. Furthermore, the architecture of the hydrophobic core at the molecular level seems to affect the occurring interactions. Block copolymer-surfactant co-assembly begins to take place at a concentration known as critical aggregation concentration (CAC—normally below the surfactants’s CMC). Besides the direct amelioration of the clinical outcome by either the active or passive targeting (EPR effect), the successful application of such smart systems offers the possibility to further protect the healthy tissues from the cytotoxic effects that most of the anticancer compounds pose. In one example the use of Pluronics, along with liquid crystalline phases based on glycerate surfactants in an injectable final formulation, achieved the controlled release of a variety of hydrophilic and hydrophobic compounds such as paclitaxel, irinotecan and octreotide. In other studies, the combination of Pluronic P123 and the nonionic surfactant Span 65 was able to formulate stable, spherical, small diameter vesicles at mild conditions in aqueous solutions and with relative ease [52,53,54]).

In general, micellar systems of block copolymers incorporating surfactants result in systems that exhibit mixed properties, and the use of multiple surfactants in a single nanosystem results in a formulation with synergistic surfactant effects [54]. An added advantage of such water-soluble systems, not uncommon to amphiphiles, is the clouding phenomenon or lower consolute temperature, which results in enhanced attractive micellar forces occurring from the increased size of polymeric aggregates leading to phase separation, with more hydrophobic compounds exhibiting a higher cloud point (CP)—phase separation temperature [16,54,55]. An interesting finding is the fact that the addition of ionic surfactants to block copolymer solutions of polyvinyl methyl ether had the exact opposite effect in terms of CP and the enhancement of the molecular polymeric attractive forces in comparison with the addition of non-ionic surfacants (which generally delay phase separation).Similar results were obtained when adding various cationic surfactants to Pluronics P84, L64, L44 solutions, as well as Reverse Pluronics 10R5, 17R4 and 25R4 [16,55]. The latter type of surfactants has the ability to create electrical charge on the micellar surface and thus generate repulsive forces, and as a result, induce hydrophilicity in the mixed polymeric system, elevating the CP [55].

While the nature and outcomes of the interaction of various surfactants of low molecular weight with block copolymers have been attracting scientific interest during the last decade, sadly, up until now, there is not a single report discussing the mixed systems of Pluronics (amongst the most widely used copolymers) with charged ionic surfactants. It should be important to elucidate the possible effects of such copolymer-surfactant interaction on the micellization process, as well as on the capacity for drug solubilization. A more recent study examined the results of the interaction between Pluronics L81, P84 and F88, with ionic surfactants having the same hydrocarbon chain lengths but different polar head groups, namely SDS, DTBA and C_12_PS. Amongst the first outcomes of the study was the conclusion that zwitterionic surfactants (C_12_PS) have a lower molecular binding affinity when compared with anionic and cationic surfactants. As a result, a higher concentration of surfactants is needed in order to decrease the final micelle size. However, regardless of the ionic nature of the surfactant used, by increasing the surfactant concentration the copolymer–surfactant mixed aggregates begin to disintegrate and, as a result, lead to micellar formations of reduced sizes, finally forming small surfactant-rich copolymer-surfactant mixed micellar nanosystems [55].

Another example of mixed Pluronic-surfactant nanosystems utilized the P85 block copolymer (M.W. = 4600) along with SDS molecules in aqueous solutions, aiming to further understand the phase transitions of the final formulations at a wide temperature window. P85 molecules transition from spherical micelles to rod-like structures at elevated temperatures, while SDS micelles exhibit an ellipsoid structure over various temperatures. The resulting mixed nanosystems showed no such phase transition, as exhibited in the block copolymer case, indicating that the lyotropism and physicochemical properties of such mixed systems can be adjusted based on the individual needs of each application by integrating different types of amphiphiles together each time [56]. 

Amongst the first observations of these type of interactions occurring between block copolymers and surfactants towards the formation of mixed aggregates via electrostatic interactions in aqueous media, was made studying a negatively charged poly(sodium methacrylate-b-ethylene oxide) copolymer (PNaMAPEO) of nearly symmetric monomer composition and single tail cationic surfactants of different chemical structure [50]. Since then, in multiple cases the creation of stable bonds between hydrophilic, neutral-ionic block copolymers and oppositely charged low molecular weight surfactants in solution, via electrostatic interactions has been examined. The resulting structures of high organization are known as complex coacervate core micelles, or C3Ms, while the most prominent example of such interaction towards the creation of internally structured surfactant/block copolymer nanoparticles is the stoichiometric mixtures of poly(acrylamide)-block- poly(acrylate) (PAAm-b-PA) along with a cationic surfactant. The resulting nanosystems had an average diameter of 50 nm without the appearance of a long-range order of organization in the lattice [57]. The binding interactions observed between anionic surfactants and uncharged polymers are much greater than the same interactions acting between uncharged polymers and cationic or neutral surfactant molecules [49,58,59]. 

While the interactions between block copolymers and surfactants have been investigated extensively over the past few decades, due to the overall complexity of the process they are still not fully understood (Figure 3). Today, it is well established that even the simplest diblock copolymer can result in a variety of well-defined, well-ordered vehicles ranging from spheres to cylinders, to other thermodynamically stable nanostructures (either 0-D, 1-D, 2-D or 3-D, while exhibiting material characteristics and phase transition properties associated with the nanoscale), depending on the relative amount, as well as on the nature and molecular weight of the components used [60]. An added advantage in terms of matter organization down to the nanoscale is that the systems that result from the complexation of such copolymers with low molecular weight surfactants exhibit hierarchical organization to various length scales and with various orientations (“structure–within–structure” morphologies with elegant nanoscopic architectural patterns). From an application point of view, the ability to transit from one structural morphology/phase to another, that the resulting mixed formulations possess, facilitates obtaining new particles without the need to repeat the whole material synthesis process. The selective removal of surfactants after the binding is complete (when possible) is also a method that can help obtain different nanostructures [60].

As previously mentioned, niosomes are amongst the most prominent nanoparticle formulations that hold great promise regarding their use as drug delivery vehicles. Niosomal formulations containing different biomaterials such as surfactants and block copolymers have been developed in multiple cases from different non-ionic surfactants. Niosomes containing the surfactants Tween 80 and Span 80 along with cholesterol with the presence of PEO-b-PCL block copolymers have shown that the presence of the polymers significantly alters the physical and structural characteristics of the niosomes, such as their size and their morphology [29]. Amongst the advantages that the above chimeric systems have to offer is their ability to encapsulate active pharmaceutical ingredients to be transferred and absorbed via multiple administration routes [29]. A key point in the use of such components in a single nanoparticulate formulation is perhaps the fact that surfactants of low molecular weight and long chain block copolymers exhibit analogous self-assembly behaviors in solutions and at interfaces, allowing for co-assembly in thermodynamically stable nanostructures [19,61]. 

A greatly studied group of macromolecular surfactants are the water-soluble triblock copolymers of poly(ethylene oxide) (PEO) and poly(propylene oxide) (PPO), widely known in the market as Pluronics, Superonics or Poloxameres (Table 3), while, according to European Pharmacopeia terms, the triblock copolymers numbered as 124, 188, 237, 338 and 407 can be described by the single use of the word Poloxamers [61]. In many instances it has been established that mixtures of polymers with surfactants exhibit enhanced properties in drug delivery applications, including formulations including the addition of more conventional non-ionic or ionic surfactants to the above block-copolymer-derived group. The incorporation of different surfactants in a single formulation exploits the possible synergistic or antagonistic interactions between them [62]. 

When using surfactants derived from different polymeric mixtures (using two or more polymer blocks of different monomeric composition), in comparison to more conventional surfactants, a key difference is that their CMC or critical micelle concentration is strongly dependent on temperature. Stable binding of small surfactant molecules and block copolymers occurs above a surfactant concentration that is below the CMC. Block copolymers of amphiphilic nature such as ExBy and ExByEx (E represents an oxyethylene unit, B represents an oxybutylene unit, and x and y denote the number of units) are surface active compounds widely known as “polymeric surfactants”. An interesting study examined the interactions of the pre-micellar region of block copolymers E58B7 and E58B11 with anionic surfactant sodium dodecyl sulfate (SDS) and cationic surfactant cetyltrimethylammonium bromide (CTMABr). It was concluded that the occurrence of particle aggregation and the overall size of the aggregates were strongly dependent on the presence of the block copolymers [63]. The results were duplicated in other studies stressing the correlation between the critical micellization temperature of Poloxamer and the surfactant content [49,64]. 

In another example, the combination of different surfactants with Poloxamer 388 (*P388*) for the encapsulation and delivery efficiency of transdermal (cream form) active pharmaceutical ingredients and use as drug delivery systems was examined. Again, the study showed that, due to the interaction of Pl388 with the surfactants present, the structure, as well as the size of the micellar aggregates, changed, while when the presence of Pl388 was more prominent—in formulations with higher polymeric concentrations—the rheological parameters of the cream bases along with the cream microstructure changed drastically, resulting in an overall more adhesive final product [65]. 

During more recent years, multiple efforts have focused on the use of nonionic block copolymers with both hydrophilic and hydrophobic monomers, especially in mixtures formed through the use of poly (oxyethylene-oxypropylene) triblock copolymers and conventional anionic, cationic or nonionic surfactants solubilized in aqueous media (Table 4). Here, an important part in understanding the effects of the underlying interactions between the different amphiphiles is the surface tension measurements that enable the detection of alterations of the original properties of the individual components [49].

While in aqueous media the multiple characteristics of the formulated nanosystems can be more easily finetuned, block copolymer/surfactant systems can create a wide range of vesicles in different organic solvents such as chloroform, many times through hydrogen bonding interactions. A characteristic example is that of a poly(styrene-b-4-vinylpyridine) block copolymer along with perfluorooctanoic acid (PFOA) that created structures of high organization and complexity in chloroform, while the triblock copolymer poly(4-vinylpyridine-b-styrene-b-4-vinylpyridine), along with the single tail non-fluorinated surfactant pentadecylphenol (PDP), have also been shown to formulate block copolymer/surfactant vesicles in organic solvents [50,56].

A category of vesicles that has been newly studied is that formulated through the interactions of block copolymers containing a polyanionic block along with surfactants of different architecture and morphologies but strictly of the opposite charge. The formulated nanoparticles, namely, block ionomer complexes (BICs), generally exhibit a hydrophobic core–hydrophilic shell structure, and are still mostly unexplored in terms of their physicochemical properties and the potential structures resulting from different combinations of cationic surfactants. Recently, a study examining the diblock copolymer PEO-*b*-PMA mixed with individual single-, double-, and triple-tail surfactants analyzed the possibility that the final morphology of the BIC nanoparticles could be altered and fine-tuned at will, based on the composition of the original mixture along with the surfactant’s original architecture, possibly allowing the introduction of such nanosystems into a broader spectrum of both industrial applications and research areas [66].

Regarding other structures besides the formulation of spherical particles as carriers, a recent study examined the physicochemical properties of a mixed anisotropic complex aggregate copolymer-surfactant system, which exhibited shape-dependent properties. The resulting formulations originated from different mixtures of PEO and -b-PNIPAm polymers along with dodecyltrimethylammonium surfactant that interacted through the development of stable electrostatic forces, ultimately resulting in a final surfactant rich–liquid crystal core and block copolymer shell in micellar cubic 3-D conformations. The selection of PNIPAm facilitated the exhibition of thermosensitivity by the nanosystems as it precipitates from water through the breakage of the hydrogen bonds at temperatures above 32 °C, exhibiting a phase transition from a sharp coil to a globule [67].

A key finding was that the desired internal organization was only achieved through the titration of the surfactant molecules to the oppositely charged polymer solution and simply mixing them with the PEO and PNIPAm molecules at the exact concentration in order to reach neutrality was not enough [67].

## 6. Characterization of Surfactant-Block Copolymer Nanosystems

The surfactant-block copolymer nanosystem is a unique class of nanosystem with several advantages for drug delivery and targeting. The preparation protocols and the techniques used for the self-assembly of surfactants and block copolymers are the conventional ones. Namely, thin-film hydration method is widely used for the preparation of these nanosystems [68]. The block copolymers and/or the surfactants are dissolved in an organic solvent and the organic solvent is evaporated under vacuum. Then, the self-assembly is achieved due to the addition of an aqueous solution or buffer [68]. Patil et al., 2014 used solvent evaporation method as preparation protocol. Supercritical antisolvent process was also used for the preparation of nanoparticles composed of block copolymers and surfactants [69,70]. Nanosuspension preparation is an alternative protocol that has appeared in the literature for the self-assembly of such systems [71].

After the preparation of the surfactant-block copolymer nanosystems, several techniques are used for their physicochemical and structural characterization. On the other hand, we should point out that before the preparation of these systems the factorial design for formulation optimization can be used in order to determine the best ratio between the components and/or the ideal physicochemical characteristics, especially size and the highest encapsulation efficiency [72]. 

After the successful preparation of the systems, a gamut of techniques is used for the elucidation of their characteristics. Firstly, dynamic and electrophoretic light scattering are used for the evaluation of the size, size distribution and zeta potential of the nanosystems. These physicochemical parameters appear in the majority of the studies because they are crucial for the route of administration, the stability, and generally for biomedical applications of the drug delivery platforms. Transmission Electron Microscopy (TEM) is also a widely used technique for the evaluation of the morphology and the shape of the systems [73]. Thermal analysis techniques, especially Differential Scanning Calorimetry (DSC), can quantify the cooperativity between surfactants and block copolymers with active ingredients. Solid or dispersion phases of the nanoparticles are subjected to heating and cooling cycles for investigating their thermotropic parameters [74]. Solubility measurements are also performed for the active ingredients in the presence of block copolymers and/or surfactants [73,74]. 

The cytotoxicity studies prove that the prepared carriers are ideal for the delivery of drugs and further biomedical applications. The drug loading efficiency has been also studied in the majority of the investigations accompanied with drug release profile curves [69,70,75]. Various analytical techniques are applied for the quantification of the loading and encapsulation efficiency of the drug. Generally, the properties of the drug molecules lead to the selection of the appropriate method. For the study of drug–excipient interactions, NMR, XRD and FTIR are also used [69,70,75].

## 7. Properties of Surfactant-Block Copolymer Nanosystems for Nanomedicine

As previously stated, multiple pharmaceutical APIs that exhibit great promise in terms of results from pre-clinical or early clinical stages show limited effectiveness when it comes to clinical practice applications. This is due to phenomena such as multi-drug resistance (often through expression of the P-glycoprotein multidrug transporter Pgp, ABCB1) and characteristics such as low aqueous solubility, poor bioavailability, short half-life, low permeability, and an unfavorable pharmacokinetic distribution inside the human organism. The administration of higher than the optimally required doses, or in some cases, continuous infusions, in order to exhibit a therapeutic response, leads to frequent toxicity problems along with high rates of unwanted side-effects, especially regarding anticancer compounds that on their own tend to exhibit high cytotoxicity [76,77]. 

The development of biodegradable carriers with active targeting capabilities holds great promise in terms of overcoming the above-mentioned issues regarding cancer treatment, improving the antitumor efficacy of already approved, clinically tested, conventional APIs [26]. Active targeting techniques help nanosized carriers overcome the size-dependency of indirect tumor accumulation through the EPR effect and thus allow formulations to achieve hydrodynamic radii above the 100–200 nm range [78]. Several antitumor compounds, including Paclitaxel, Camptothecin, Harmine, Docetaxel, Myricetin, Gambogic acid, Methotrexate, iCariside, Genisten, as well as antioxidant, plant-extracted curcumin, have been linked with the internal hydrophobic cores of various mixed micellar nanoformulations towards the formation of nanocarriers of relatively wide size ranges (50 nm–450 nm). All of them have been found to exhibit increased drug permeability and bioavailability, lower toxicity levels through lower administered doses (with better elimination profiles), while, most importantly, all of which achieved site-specific distribution and accumulation [73,76,77,79,80,81,82,83,84]. 

Recently, Pluronic F87-poly(lactic acid) (FA-F87-PLA) micelles, along with vitamin E TPGS, were combined with FA-F87-PLA to formulate advanced micellar formulations aimed at improving the intravenous delivery, cellular uptake and efficacy of Paclitaxel (PTX). The formulation showed average sizes of 50 nm to 94 nm which, interestingly, were smaller than the same nanoparticles without vitamin E (101 nm), indicating that up to a certain concentration the addition of TPGS resulted in a decreased particle radius and increased LE, possibly resulting in a higher cellular uptake (exhibited mostly during the first 2 h and 4 h after initial incubation), while also strongly stressing vitamin E’s effectiveness as an emulsifier. Follow-up in vivo pharmacokinetic studies demonstrated that the PTX-loaded FA-F87-PLA/TPGS mixed micelles had an AUC of almost 1.4 times higher in comparison to that of PTX-loaded FA-F87-PLA micelles [73]. 

The utilization of solubilizing nanocarriers specifically designed for the delivery of paclitaxel, aiming for its successful incorporation in today’s clinical practice, is not unprecedented. In most cases the use of the Pluronic triblock copolymer is a pre-requisite for a successful polymeric formulation, along with a variety of other biodegradable molecules, surfactants and solubilizers. Recently, paclitaxel-loaded nanoparticles were synthesized using Pluronic copolymers (F-68 and P-123) along with the non-ionic surfactant Span 40 as a nanocarrier. In terms of the release profile of the formulations, studies showed that the total amount of paclitaxel released during a 96 h time period was around 65% of the original encapsulated drug, while the overall drug efficiency was better during the first 48 to 72 h when compared with the administration of the original pharmaceutical product [77].

Studies conducted using cervical cancer Hela cells exhibited that a dose of 5 ng/mL PTX loaded in the nanoparticles resulted in the death of more than 80.23% of cancer cells, whereas the same free-dose resulted in the death of no more than 38.49% of the cells during the first 72 h of exposure (*p* < 0.05), while in breast cancer MCF-7 cells, 5 ng/mL PTX-NPs and PTX induced a 63.95% and 47.45% cell death rate at 72 h, respectively [77]. Lastly, in comparison with control cells in only 48 h the paclitaxel-loaded nanoparticles achieved a decrease in cell viability by 50%. The overall toxicity of the PTX-NPs was also significantly lower when compared to the use of paclitaxel in its original form [77].

It is worth stressing that the successful incorporation of vitamin E in a mixed micellar nanoformulation for the amelioration of the pharmacokinetic characteristics and bioavailability of an anticancer compound was also demonstrated during the encapsulation of docetaxel (DTX) for per os delivery. The in vitro release study indicated that DTX-loaded MPP/TPGS/CSO-SA MPMs exhibited significantly slower release rates when compared to DTX solution, while at the same time, the oral bioavailability of the DTX-loaded mixed micelles was 2.52 times higher in comparison to that of the DTX solution alone [79]. 

Another example where the enhancement of the effectiveness of an anticancer compound, as well as the need to overcome multi-drug resistance due to pump P-gp efflux, was the overall aim, is that of the use of mixed micelles for the oral transportation of Harmine (HM), a BCS class II compound. HM-loaded galactosylated pluronic F68-gelucire 44/14 mixed micelles achieved a six-fold increase in HM when compared with the administration of Harmine alone, while achieving a seven-fold increase in the nanoparticle presence in the site of interest (liver cancer cells). The overall sizes of the mixed micelles ranged between 277 nm and 595 nm, which in turn was strongly correlated with the concentration of Pluronic in the final nanoformulation. Lastly, the mass of the actually encapsulated HM through the various formulations was found to range from 60.5% to 89.9% *w*/*w*, indicating that the above nanosystem could be considered as an attractive choice for the encapsulation of BSC II category drugs, offering protection from rapid elimination and, at the same time, higher blood circulation times, due to the hydrophilic outer shell and its overall small size [76].

Other studies, utilized mixed systems composed of block copolymers along with micelles, aiming for the successful delivery of various APIs to affected brain cells, either directly through providing the nanoformulation with the necessary tools to overcome the blood brain barrier, or indirectly, via fine-tuning certain physicochemical characteristics that will enable the system to prolong its blood circulation time and improve its bioavailability, increasing in turn the chances that eventually it will reach the cells of interest [72,74,80].

In nanomedicine, the route of administration is amongst the most prominent factors that affect the nanoparticles’—and by extension the active substance’s—ability to reach the area of interest, while retaining their structure, size and characteristics intact, and, more importantly, their ability to be absorbed and distributed as intended. The microenvironment with which the nanoparticles are first in contact contains its own unique biological barriers (extracellular and endocellular) and mechanisms of recognition and clearance, different proteins (formation of NP-protein corona) and macromolecules that might be in abundance, and even different pH. 

Utilizing the knowledge that the route of administration strongly affects the drug’s overall efficiency, mixed polymeric micelles composed of Gelucire 44/14 and Pluronic F127 were developed for the successful delivery of lurasidone HCl (LH) to the brain via nasal administration, a neurotherapeutic molecule indicated for managing the symptoms of bipolar disorder and schizophrenia [72]. Nasal administration offers the advantage of being a non-invasive technique that can —depending on the pathway (Figure 4) that the nanoformulations choose after administration—presumably surpass the blood brain barrier [72]. Stable NP–Lurasidone formulations achieving an average size of 175 nm exhibited an overall 93.13 ± 0.08 to 98.35 ± 0.01% drug entrapment efficiency, increased permeation through the nasal mucosa (79 ± 0.02% drug permeation in contrast to 59 ± 0.12% from plain drug suspension after 10 h), while also achieving a controlled drug release profile, thus resulting in a better pharmacokinetic and drug—plasma levels profile [72].

The final nanoparticles were spherical in shape, and in vivo pharmacokinetic studies in rats stressed the superiority of brain targeting through nasal administration in contrast to intravenous. The use of sheep nasal mucosa that remained intact after the end of the treatment exhibited the safety of the nanoformulation [72].

In another study, lipid-based mixed micelles, along with sodium dodecyl sulphate (SDS), Pluronic F68 (F68) and Labrasol, were formulated for the encapsulation and delivery of myricetin (MYR) to glioblastoma cells (U251) through oral delivery. The use of non-ionic surfactants aimed to create a protective exterior coating layer to protect the micelles, and thus myricetin, from oxidative degradation, achieving an overall preferable uptake from brain cells when compared with the free drug, while also achieving an adequate myricetin loading [80]. The final nanocarriers ranged in size from 54 nm up to 139 nm, while when tested in a more acidic environment simulating that of our GI tract (pH 6.0 to pH 2.0) the release of MYR from the NPs was enhanced [80].

In terms of distribution, excretory organs such as the liver and kidneys exhibited a lower nanoparticle—drug concentration when compared with the free drug, while the retention from the targeted cells was higher, resulting in a prolonged and more pronounced myricetin presence, leading to an augmented cell death rate [80].

Lastly, Pluronic/phosphatidylcholine/polysorbate 80 mixed micelles (PPPMM) were created recently for the encapsulation of Nimodipine (NM), an FDA approved medication aimed at the treatment of subarachnoid hemorrhage induced vasospasm, with the ideal formulations consisting of 75:25 molar ratios of phosphatidylcholine to Pluronic. The mixture resulted in formulations with large, hydrophobic cores, and thus enabled the maximum drug encapsulation (1.06 ± 0.03 mg/mL) and resulted in great plasma and brain drug bioavailability exhibited in vivo using rat models (bioavailability in plasma 232% and brain 208%). The current study indicates the importance of examining the incorporation of phosphatidylcholine in stably linked mixed micelle formulations—in this case P123/P127 mixed micelles—acting as an enhancer of per os absorption and bioavailability by drastically improving portal blood absorption and lymphatic delivery [74].

The use of P127, along with more conventional surfactant molecules in mixed micellar nanoformulations, for the entrapment of low solubility APIs can be witnessed in several studies, indicating its importance as an effective solubilizer amongst non-ionic surfactants of larger molecular weight. Isolated from Maclura pomifera, morin is a chemical compound (flavonoid, anticancer, anti-inflamatory, antioxidant) used to inhibit fatty acid synthase and amyloid formation by islet amyloid polypeptide, and was encapsulated into Pluronic F127–Tween 80 mixed micelles (1:10 *w*/*w* ratio), with the total mass of the encapsulated morin being 0.02 of that of the Tween 80 surfactant. In vivo rat experiments indicated an increase of oral morin bioavailability reaching 11.2% in comparison with free morin digestion, limiting the bioavailable percentage of the administered dose to 0.4%. Besides the low solubility, these levels are attributed to the low permeability of the active compound through the intestinal mucosa, that due to the first pass effect, only a small quantity of which can pass from the acidic stomach environment to the small intestine [68]. 

Another compound with antioxidant, anti-inflammatory and anticancer properties similar to morin is resveratrol, also exhibiting similar limitations in terms of solubility, bioavailability and permeability. In vitro studies in Caco-2 cells of mixed nanocarriers consisting of Soluplus (polymer) and Tween 80 in molar ratios 2:1 exhibited a 2000-fold enhancement of drug solubility, while follow-up in vivo experiments of reservatol:Soluplus–P 407 (1:2–15%) mixed carriers in solid dispersion form presented an AUCo-t of 279 ± 54 ng.h/mL and a C_max_ of 134 ± 78 ng/mL, a 2.5-fold increase when compared to solid dispersions without the use of Poloxamer 407. Besides the increase in overall solubility, it was most likely the reduction of intestinal efflux (efflux transporters in the membranes of the intestine determining drug deposition in the organism), as well as the activity of specific metabolism mechanisms, that led to rapid drug clearance and facilitated resveratrol’s increased final bioavailability [75]. 

Solid nanoparticle dispersions, with and without the presence of surfactants have also been examined for the enhancement of the solubility and bioavailability of celecoxib, an anti-inflammatory active compound used for the treatment of arthritis-related illnesses. The study resulted in spherical-shaped celecoxib nanoparticles of average sizes of 300 nm in solid dispersion, along with polyvinylpyrrolidone (PVP) (2:8 ratio), while other surfactants such as gelucire 44/14, poloxamer 188, poloxamer 407, Ryoto sugar ester L1695, and D-α-tocopheryl polyethylene glycol 1000 succinate (TPGS) were evaluated in terms of altering the NP’s morphological, rheological and physicochemical characteristics. In terms of oral bioavailability, the celevoxib-PVP-TPGS nanoformulations achieved an in vitro dissolution and absorption comparable with that of the free drug, while in terms of the AUC concentration, the first 24 h after initial administration, as well as the C_max_, exhibited an increase of 4.6 to 5.7 times, respectively [70].

## 8. Challenges and Future Perspectives

Every year, more and more active pharmaceutical compounds and drug products that are indicated for the treatment of severe diseases such as cancer, genetic diseases and autoimmune disorders exhibit properties such as low solubility and bioavailability, along with high lipophilicity (BSC Class II & IV drugs), resulting in therapeutic protocols that require the administration of higher than optimal doses in order to achieve the desired clinical outcome. Moreover, the administration of therapeutic compounds in areas and tissues of the organism of low accessibility (i.e., CNS conditions, blood–brain barrier transcendence, subcellular targeting, etc.) require the development of novel approaches in terms of drug encapsulation and delivery with minimal toxicity and higher tissue-specific targeting. To that end, nanomedicinal drug delivery platforms that combine the advantages of multiple innovative and low toxicity materials, such as polymers and surfactants, are being extensively studied, exhibiting promising preclinical results towards the treatment of various disorders. Such amphiphilic in nature molecules have the ability to self-assemble in aqueous media formulating nanoscale systems of high organization with well-defined morphologies, offering the possibility to fine-tune and alter their physicochemical characteristics with respect to each application. The formation of core-shell structures such as micelles, resulting from the combination of such molecules, enables the incorporation of both hydrophobic and hydrophilic drugs, while the surface functionalization of such (targeting moieties, PEGylation, monoclonal antibodies etc), allows for the attribution of additional advantages, such as microenvironment identification, prolonged circulation times and active targeting capability. Due to their increasing popularity worldwide, numerous researchers, institutes, and pharmaceutical companies are continuously working to solve the problems associated with scaling, characterization, distribution, and safe clinical use in everyday life, and due to their progress, regulatory agencies are now creating the necessary guidelines that will allow for easier approval of such products so that they can more quickly begin to penetrate global markets.

Despite the promising preclinical results that such delivery platforms exhibit however, there are still great strides to be made towards their successful incorporation in today’s clinical practice. It has been highly stressed that future success is dependent upon the correct choice of polymers (non-toxic) and chimeric particles, regarding their proposed use, the administration route, the emergence of novel APIs, as well the evolution of the already existing analytical characterization techniques, towards more cost-effective and simpler to interpret assays. Proper characterization of such nanosystems requires the use of highly expensive techniques (i.e., cryo-TEM, thermal analysis techniques etc.), while batch-to-batch variability also presents a significant roadblock with respect to production scaling. Lastly, the majority of such nanoscale delivery platforms fail to proceed beyond Phase II of clinical trials, either due to rising toxicity issues or due to lower than previously exhibited (Phase I) therapeutic efficacy. It is our belief that systems of a mixed nature, such as the ones discussed in the length of this manuscript, represent the best efforts in utilizing the tools of nanotechnology in drug delivery applications, being able to overcome the problems associated with the use of more conventional drug products. In any case, as with any other known drug, the uptake of a nanomedicinal product into clinical practice depends on the balance between efficacy and safety.

## 9. Conclusions

This review has examined what is currently known and recently discovered regarding polymeric and mixed block copolymer-surfactant based nanoparticulate systems for nanotechnological applications in the field of medicine aimed at the treatment of various disorders. Each section has highlighted different categories of nanosystems, presenting a holistic view of those, starting from their development process and their characteristics, to real-life applications of such. Different approaches, both in vitro and in vivo, their advantages and shortcomings, as well as their probable future applications, have been discussed in depth. Significant advances in the field are expected in the near future due to the interesting properties provided by the mixed nanosystems and their variability in composition, morphology, and nanoscopic encapsulation and delivery properties as a result of the availability of the many chemically different components that can be utilized and fruitfully co-assembles in functional nanostructures.

## Figures and Tables

**Figure 1 pharmaceutics-15-00501-f001:**
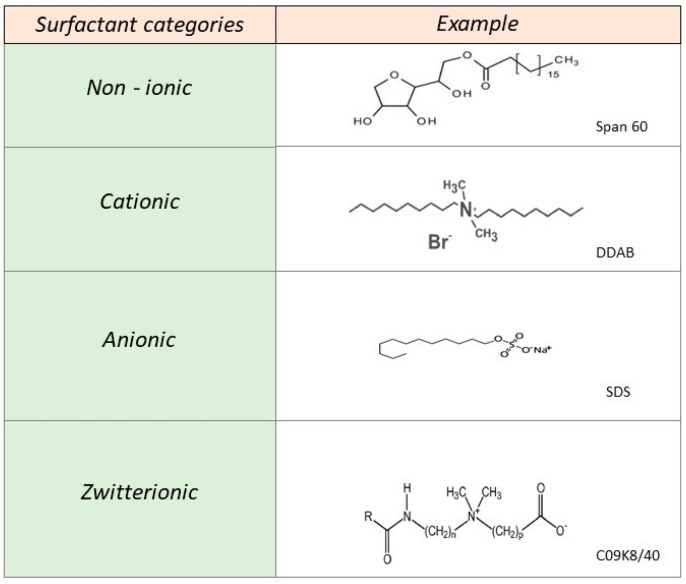
Surfactant classification based on their charge and examples of chemical structures.

**Figure 2 pharmaceutics-15-00501-f002:**
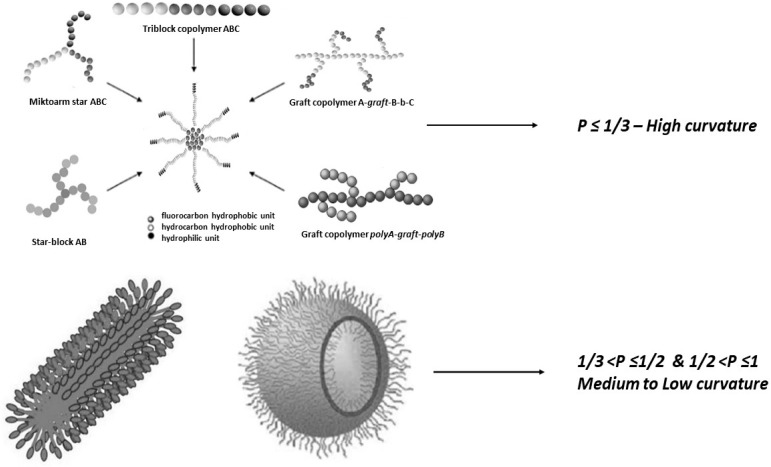
Block copolymer self-assembled nanostructures based on the hydrophilic/lipophilic balance of individual chain polymers, ranging from spherical and cylindrical micelles to vesicles.

**Figure 3 pharmaceutics-15-00501-f003:**
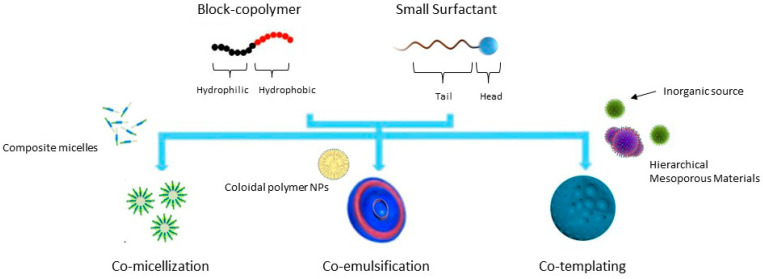
Binary amphiphilic co-assembly of amphiphilic block-copolymers with small weight surfactants.

**Figure 4 pharmaceutics-15-00501-f004:**
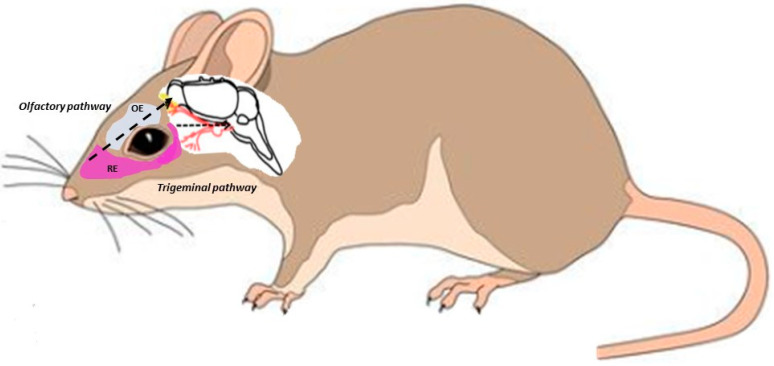
Intranasal pathways for brain absorption of nanoformulations.

**Table 1 pharmaceutics-15-00501-t001:** Surfactants mostly used in drug delivery.

Surfactant	Category	HLB	Groups
Span 85	Non-ionic	1.8	Hydrophobic (oil soluble)
Span 65	Non-ionic	2	Hydrophobic (oil soluble)
Span 80	Non-ionic	4.3	Hydrophobic (oil soluble)
Span 60	Non-ionic	4.7	Hydrophobic (oil soluble)
Span 40	Non-ionic	6.7	Hydrophobic (oil soluble)
Span 20	Non-ionic	8.6	Water Dispersable
Brij 30	Non-ionic	9.5	Water Dispersable
Tween 61	Non-ionic	9.6	Water Dispersable
Tween 81	Non-ionic	10	Water Dispersable
Tween 65	Non-ionic	10.5	Water Dispersable
Tween 85	Non-ionic	11	Water Dispersable
PEG 400 monooleate	Non-ionic	11.4	Water Dispersable
PEG 400 monostearate	Non-ionic	11.6	Water Dispersable
Brij 97	Non-ionic	12.5	Hydrophilic (water soluble)
Lipocol C-10	Non-ionic	12.9	Hydrophilic (water soluble)
Tween 21	Non-ionic	13.3	Hydrophilic (water soluble)
Tween 60	Non-ionic	14.9	Hydrophilic (water soluble)
Tween 80	Non-ionic	14.9	Hydrophilic (water soluble)
Tween 40	Non-ionic	15.6	Hydrophilic (water soluble)
Tween 20	Non-ionic	16.7	Hydrophilic (water soluble)
Brij 35	Non-ionic	16.9	Hydrophilic (water soluble)
Solutol SH-15	Non-ionic	15.2	Hydrophilic (water soluble)

**Table 2 pharmaceutics-15-00501-t002:** Examples of nanoparticle surfactant drug delivery systems.

Drug	Surfactant	Method of Preparation	Delivery	Nanosystem	Reference
Methotrexate	Span 60	Thin film hydration	Transdermal	Niosomes	Abdelbary AA, AbouGhaly MHH. Int J Pharm. 2015; 485(1–2): 235–43 [22].
Risperidone	Tween 20, 60, 80 & Span 20, 40, 60, 80	Proniosome derived niosome method	Transdermal	Niosomes	Sambhakar S, Paliwal S, Sharma S, Singh B. Bull Fac Pharm Cairo Univ. 2017; 55(2): 239–47 [23].
Ciprofloxacin	Span 60, Tween 60	Thin film hydration	Pulmonary	Niosomes	Moazeni E, Gilani K, Sotoudegan F, Pardakhty A, Najafabadi AR, Ghalandari R, et al. J Microencapsul. 2010; 27(7): 618–27 [24].
Cisplatin	Span 40	Emultion method	Parenteral	Niosomes	Yang H, Deng A, Zhang J, Wang J, Lu B. J Microencapsul. 2013; 30(3): 237–44 [25].
Docetaxel	Pluronic F127, Span 80	High-energy method	Subcutaneous	SLNs	da Rocha MCO, da Silva PB, Radicchi MA, Andrade BYG, de Oliveira JV, Venus T, et al. J Nanobiotechnology [Internet]. 2020; 18(1): 43 [21].

**Table 3 pharmaceutics-15-00501-t003:** Pluronic block copolymers with various chain architecture characteristics.

Pluronic^®^	No. of EO Units	No. of PO Units	Mass	HLB	CP in 1% Aqueous Solution (°C)
L61	4.55	31.03	2000	3	24
F127	200.45	65.17	12,600	22	>100
F68	152.73	28.97	8400	29	>100
F87	122.5	39.83	7700	24	>100

**Table 4 pharmaceutics-15-00501-t004:** Block-copolymer/surfactant-based formulations for innovative drug delivery applications.

Block-Copolymer	Surfactant	Drug	Use
PDLLA	mPEG	Paclitaxel	Breast cancer
d,l-lactide, glycolide, ε-caprolactone	Pluronic F-68	Potentially protein drugs	Injectable, sustained release formulation
ABA, A(BA)_n_, or B(AB)_n_ hydrophilic/hydrophobic block-copolymers	PEG	Labile peptide and protein drugs, antibiotics, adriamycin, mitomycin, bleomycin, cisplatin, carboplatin, doxorubicin, daunorubicin, 5-fluorouracil, methotrexate, taxol, taxotere, actinomycin D	Anticancer, antimicrobial, anti-inflamatory
polycaprolactone (PCL)	Pluronic F-68	indocyanine green (ICG)	Imaging and photothermal therapy (PTT)

## Data Availability

Not applicable.

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
