# Peer review of "Surfactant and Block Copolymer Nanostructures: From Design and Development to Nanomedicine Preclinical Studies"

_pharmaceutics, 2023, doi:10.3390/pharmaceutics15020501_

Round 1

Reviewer 1 Report

In the present review article titled “Surfactant and block copolymer nanostructures: from design and development to nanomedicine preclinical studies”, the authors described categorize, summarize and present, selected recent scientific results on the applications of surfactant based, polymer based and mixed, nanoparticulate drug formulations intended for use in different ways in the medical field and in drug delivery

According to my point of view, this aim and rationale of the review was already discussed in many literatures, may be in different manner, but already reviewed before and most basic information are present in any text book. I do not find any novelty throughout the manuscript, more effort need to be done in data collection and amalgamation.

The following points should be taken in consideration and be revised.

·         References need to be updated, only 3 references in 2022 and 7 references in 2021.

·         Figures quality is poor. Figure 6 is not accepted at all for me.

·         Authors can add more examples and collect data in tables to be clearer to readers.

·         Materials and Methods section should be omitted, I have not seen that section before when dealing with review article as a state of art.

·         Where is the expert opinion of the authors and their comment on each section regarding the core of the review

Author Response

Reviewer: 1

Comment

According to my point of view, this aim and rationale of the review was already discussed in many literatures, may be in different manner, but already reviewed before and most basic information are present in any text book. I do not find any novelty throughout the manuscript, more effort need to be done in data collection and amalgamation.

Answer

The manuscript presented, focuses on the nature, properties and characterization of polymer nanosystems, surfactant nanosystems and, more importantly, block-copolymer surfactant mixed systems regarding mainly the application of such innovative nanosystems in the field of nanomedicine. The information provided, while in certain instances can be found fragmentally in other sources published in esteemed peer review scientific journals and books, is presented in a structured, comprehensive and wholistic manner regarding all materials used, something that is absent in the international literature.  Moreover, the focus of this work is to examine the most prominent developments regarding the application of such material as components of novel drug delivery systems that exhibit targeting capabilities, stimuli-responsiveness and microenvironment identification, while at the same time stressing the role of their physicochemical, morphological and thermodynamic characteristics (for example self-assembly driving forces, phase transition properties etc).

Comment

 References need to be updated, only 3 references in 2022 and 7 references in 2021.

Answer

Regarding the reviewer’s comment, a systematic search was made in multiple databases (Pubmed, MedLine, Scopus, Google Scholar, Pergamos etc), during which process, research works that did not meet our selection criteria (published without prior peer review, duplicate records, insufficient details, out of scope etc) were excluded. The aim was to gather and extrapolate as much data as possible to provide a holistic, spherical view regarding our research goal. The following combination of terms was used to identify and collect the information required for the first phase as previously described: polymeric nanosystems, drug delivery, block-copolymer nanosystems, surfactants, non-ionic surfactants, ionic surfactants, block-copolymer self-assembly, niosomes, pH responsiveness, polymeric nanoparticles, RAFT polymerization, Pluronic, Tween, Brij, Span etc. Regarding the results of such research and our exclusion process, we concluded in utilizing the information provided in more than 45 papers published after 2016.

Moreover, the following citations were added into the manuscript:

Yu, J., Qiu, H., Yin, S., Wang, H., & Li, Y. (2021). Polymeric drug delivery system based on pluronics for cancer treatment. Molecules (Basel, Switzerland), 26(12), 3610. doi:10.3390/molecules26123610.

Raval, N., Kalyane, D., Maheshwari, R., & Tekade, R. K. (2019). Copolymers and block copolymers in drug delivery and therapy. In R. K. Tekade (Ed.), Basic Fundamentals of Drug Delivery (pp. 173–201). San Diego, CA: Elsevier.

Comment

Figures quality is poor. Figure 6 is not accepted at all for me.

Answer

We would like to thank the reviewer for bringing the particular concern to our attention. As suggested, the quality of all the figures has been enhanced.

Concerning the reviewer’s comment for Figure 6 (as per new numbering Figure 4), we would like to stress, that an important aspect of every drug delivery product (innovative or not) with respect from its formulation process and design to its clinical effectiveness, rests with the route of drug administration. Different pathways can ameliorate drug effectiveness by helping transcend certain biological extracellular and intracellular barriers, while also have a direct effect on other pharmacokinetic and toxicological aspects. Figure 4 in the present version stands as a clear example of such phenomena in applications of nose to brain delivery, as stressed in the paragraph above it, offering a graphical representation of a novel, non-invasive delivery platform, while exhibiting the drug route towards the brain.

Comment

Authors can add more examples and collect data in tables to be clearer to readers.

Answer

With respect to the reviewer’s comment, “Table 4. Block-copolymer/surfactant-based formulations for innovative drug delivery applications” was added into the manuscript, in section 5. “Surfactants-block copolymer mixed nanosystems”.

Comment

Materials and Methodsection should be omitted, I have not seen that section before when dealing with review article as a state of art.

Answer

With respect to the reviewer’s comment, we provide the following review articles published at multiple MDPI journals that contain the section Materials and Methods or analogous ones. The provision of this section is essential in order to explain the origin and manner by which the information provided in a review article were gathered and processed by the authors. Furthermore, we have proceeded in renaming this section to Methodology.

Bernsen EC, Hogenes VJ, Nuijen B, Hanff LM, Huitema ADR, Diekstra MHM. Practical recommendations for the manipulation of kinase inhibitor formulations to age-appropriate dosage forms. Pharmaceutics. 2022;14(12):2834.

Shi P, Tang Y, Zhang Z, Feng X, Li C. Effect of physical exercise in real-world settings on executive function of typical children and adolescents: A systematic review. Brain Sci. 2022;12(12):1734.

Alhjouj A, Bonoli A, Zamorano M. A critical perspective and inclusive analysis of sustainable road infrastructure literature. Appl Sci (Basel). 2022;12(24):12996.

Wieruszewski M, Mydlarz K. The potential of the bioenergy market in the European Union—an overview of energy biomass resources. Energies. 2022;15(24):9601.

Comment

Where is the expert opinion of the authors and their comment on each section regarding the core of the review.

Answer

With respect to the reviewer’s comment, the following section “8. Challenges and Future Perspectives” has been added in the manuscript, discussing the authors views in terms of the current state of block-copolymer/surfactant nanosystems in today preclinical and clinical practice. The promise/potential that such systems exhibit, as well as possible short-comings, and issues that need to be addressed are discussed in this section.

Reviewer 2 Report

The main idea of the review manuscript “Surfactant and block copolymer nanostructures: from design and development to nanomedicine preclinical studies” is interesting.  The manuscript is well written. I suggest to publish in the current state.

Author Response

We would like to thank you very much for your comments.

Reviewer 3 Report

The manuscript reviewed the surfactants and block-coploymers and both for drug delivering, but the contents were arranged in confusing style for readers difficult understanding and reading. I suggest the authors rearranged  surfactants according to non-ionic, cationic, anionic,zwitterionic; and block-polymers to linear copolymers, branching or star copolymers as mentioned in the manuscript. In one word, The contents nend ro be rearranged for easy reading and understanding.

Author Response

Comment

 I suggest the authors rearranged surfactants according to non-ionic, cationic, anionic, zwitterionic; and block-polymers to linear copolymers, branching or star copolymers as mentioned in the manuscript. In one word, The contents nend to be rearranged for easy reading and understanding.

Answer

We would like to thank the reviewer for bringing the particular concern to our attention. As suggested, the contents of the manuscript have been rearranged accordingly.

Round 2

Reviewer 3 Report

I have no more comment on the revised manuscript.